# Artificial intelligence model for analyzing colonic endoscopy images to detect changes associated with irritable bowel syndrome

**Kazuhisa Tabata, Hiroshi Mihara◉\*, Sohachi Nanjo, Iori Motoo, Takayuki Ando, Akira Teramoto◉, Haruka Fujinami, Ichiro Yasuda**

3rd Department of Internal Medicine, Graduate School of Medicine, University of Toyama, Toyama, Toyama, Japan

\* m164.tym@gmail.com

**Data Availability Statement:** All image datasets used in this study can be found at https://doi.org/ 10.5061/dryad.9s4mw6mkp. Several images were

## Abstract

IBS is not considered to be an organic disease and usually shows no abnormality on lower gastrointestinal endoscopy, although biofilm formation, dysbiosis, and histological microinflammation have recently been reported in patients with IBS. In this study, we investigated whether an artificial intelligence (AI) colorectal image model can identify minute endoscopic changes, which cannot typically be detected by human investigators, that are associated with IBS. Study subjects were identified based on electronic medical records and categorized as IBS (Group I; n = 11), IBS with predominant constipation (IBS-C; Group C; n = 12), and IBS with predominant diarrhea (IBS-D; Group D; n = 12). The study subjects had no other diseases. Colonoscopy images from IBS patients and from asymptomatic healthy subjects (Group N; n = 88) were obtained. Google Cloud Platform AutoML Vision (single-label classification) was used to construct AI image models to calculate sensitivity, specificity, predictive value, and AUC. A total of 2479, 382, 538, and 484 images were randomly selected for Groups N, I, C and D, respectively. The AUC of the model discriminating between Group N and I was 0.95. Sensitivity, specificity, positive predictive value, and negative predictive value of Group I detection were 30.8%, 97.6%, 66.7%, and 90.2%, respectively. The overall AUC of the model discriminating between Groups N, C, and D was 0.83; sensitivity, specificity, and positive predictive value of Group N were 87.5%, 46.2%, and 79.9%, respectively. Using the image AI model, colonoscopy images of IBS could be discriminated from healthy subjects at AUC 0.95. Prospective studies are needed to further validate whether this externally validated model has similar diagnostic capabilities at other facilities and whether it can be used to determine treatment efficacy.

## Author summary

This study reports on an endoscopic image artificial intelligence (AI) model for detecting irritable bowel syndrome (IBS). Endoscopic images of IBS patients usually do not have any teacher data because their changes cannot be detected by a human observer. However, we investigated the possibility of using the presence or absence of symptoms as teacher

removed from the repository as the data contain potentially sensitive information such as a partial patient ID in the image.

**Funding:** The authors received no specific funding for this work.

**Competing interests:** The authors have declared that no competing interests exist.

data, and found that endoscopic images of IBS patients could be discriminated with high accuracy from those of healthy subjects, and that endoscopic images of diarrhea-type IBS could also be discriminated from those of constipation-type IBS. It is expected that this will enable endoscopic AI diagnosis in other functional gastrointestinal diseases such as NERD and functional dyspepsia by building an image AI model based on the presence or absence of symptoms. In addition, this study uses a code-free deep learning approach, which has the potential to improve clinicians' access to deep learning. Further research is needed to determine whether real-time IBS image determination as well as prediction of treatment efficacy is possible.

## Introduction

Irritable bowel syndrome (IBS) affects about 10% of the Western population, and its prevalence is increasing annually [1]. Patients with IBS frequently experience abdominal pain and changes in stool habits, but often exhibit no abnormalities in immediate diagnostic tests or lower gastrointestinal endoscopy [2]. Recent evidence indicates that aspects of Western lifestyles, such as frequent antibiotic therapy, that alter the microbiota, may be involved in development of IBS. Biofilm formation is a unique mode of microbial growth [3] and polymicrobial biofilms have been implicated in several gastrointestinal disorders [4,5]. In a recent study, biofilm formation by *E. coli* and *Ruminococcus gnavus* in the terminal ileum to the ascending colon was seen in 60% of IBS cases, and close observation revealed changes in areas where biofilms formed [6]. However, even though microinflammation is present histologically, human investigators currently cannot determine whether an endoscopic image is from a patient with IBS.

Imaging artificial intelligence models (AI) have been developed to detect lower gastrointestinal tract lesions in real time, and several models have already been clinically applied [7]. Complex AI models such as those developed for imaging applications require deep learning algorithms and generally can only be built using Python libraries and require programming expertise. Although relatively few physicians have such skills, tools such as Google Cloud Platform (Google Inc. Mountain View, CA Available at: http://cloud.google.com/vision/. Accessed 13 Feb 2022) now allow the building of AI models without such programming expertise, and thus the application of AI models in the medical field is likely to expand [8,9]. In fact, use of AI in the pathological diagnosis of infertile spermatozoa and otolaryngology imaging has already been reported [10,11].

Typically, training datasets are needed to develop AI, but such datasets are not available for functional gastrointestinal diseases, which do not show abnormalities even on endoscopy. However, inclusion of additional information such as the presence or absence of symptoms in training datasets used to develop AI models may allow detection of minute changes in the colon that cannot be detected by human observers. The purpose of this study was to determine whether AI models for image analysis can differentiate between different types of IBS and healthy colonoscopic images in real-world clinical practice using Google cloud AutoML Vision.

## Materials and methods

### Ethics

The study protocol was approved by the Ethics Committee of the Toyama University hospital (approval No. R2021032). All methods were performed in accordance with the relevant guidelines and regulations as well as with the Declaration of Helsinki. The study design was accepted

by the ethics committee on the condition that a document declaring an opt-out policy (the need for consent was waived by the ethics committee) by which any potential patients and/or their relatives could refuse to be included, was uploaded to the website of the Toyama University hospital.

For use in real-world patients with IBS, patients were identified not by ROME criteria, but instead based on disease names recorded for insurance purposes between January 2010 and December 2020. These names included "Irritable bowel syndrome" (Group I), "constipated irritable bowel syndrome" (Group C), and "diarrhea irritable bowel syndrome" (Group D). Other diseases such as colorectal cancer, inflammatory bowel disease, and eosinophilic gastroenteritis were excluded based on symptoms and results of histopathological examinations. However, cases with nonspecific inflammatory cell infiltrates that did not meet the diagnostic criteria and were being followed up under the respective insurance disease names were included in the relevant group. For symptomatic patients, colonoscopy was done as part of a workup for changes in bowel habits (e.g., diarrhea), and asymptomatic patients had undergone colorectal cancer screening. Asymptomatic patients comprised Group N. Colonoscopy images were obtained from the endoscopy reporting system. Images were taken by more than 10 trainees or specialists at a single institution with an Olympus CF-HQ290Z or PCF-H290Z colonoscope. The accuracy of the model was improved by building the model multiple times after excluding normal light images of the terminal ileum, rectal inversion and anus, narrow band or dye-spread images. No biofilm was detected. Images having scores of 2 (i.e., minor amount of residual staining, small fragments of stool and/or opaque liquid, with well-visualized mucosa of colon segments) and 3 (entire mucosa of the colon segment was well-visualized with no residual staining, small fragments of stool and/or opaque liquid) on the Boston Bowel Preparation Scale (BBPS) [12] were employed. A total of 20 to 40 images were used, with about 5 images for regions in each segment (cecum, ascending, transverse, descending, sigmoid colons and rectum) per patient. Groups N, I, C, and D had 88, 11, 12 and 12 patients, respectively, for which 2,479, 382, 538, and 484 images, respectively, were used. The accuracy of the model increases with the number of patients, but a minimum of 100 images afforded a certain degree of accuracy. Thus, the number of patients and images used was considered sufficient to construct this model.

In this study we used annotation and Algorithm Generation using Google Cloud AutoML Vision from the Google Cloud Platform (GCP) (Google, Inc.). Four labels were defined as Groups N, I, C and D in the training dataset (single label classification). Three models were produced that differentiated Group I and N, Group N, C and D, or Group C and D. This process was done entirely by a single physician (HM).

## Artificial neural network programming, training and external validation

The Google Cloud AutoML Vision platform was used to automatically and randomly select training set images (80%), validation set images (10%), and test set images (10%) from the dataset for algorithm training processes. Since the images are independent, an external validation can be performed. A total of 16 nodes (2 hours) were used to train the algorithm. AutoML Vision provides metrics: positive predictive values and sensitivity to stated thresholds, and area under the curve (AUC). For each model, we also generated a confusion matrix that cross-references the true labels against the labels predicted by the deep learning mode [8]. Using the extracted binary diagnostic accuracy data, we created a contingency table (confusion matrix) showing the calculated values for specificity at a threshold of 0.5. The confusion matrix showed results for true positive, false positive, true negative, and false negative. The probability of a given label for each image is presented as a score between 0 and 1.

## Results

### Group N vs. Group I

The first question we addressed was whether AI could distinguish patients with irritable bowel syndrome from healthy subjects. IBS is classified as IBS-C, IBS-D, and IBS-MIX, but their percentage in Group I has not been determined. In Japan, ramosetron and linaclotide are covered for IBS-D and IBS-C, respectively, resulting in Group D being patients prescribed ramosetron, Group C being patients prescribed linaclotide, and Group I being IBS patients not prescribed either. As training, validation and test images, 1969, 255 and 255 images, respectively, were used for Group N and 304, 39 and 39 images, respectively, were used for Group I. A comparison of patients in Group N for whom endoscopy showed no apparent abnormalities and Group I showed that the average precision (positive predictive value), precision and recall of the algorithm was 94.6%, 88.78% and 88.78%, respectively, based on automated training and testing using the AI model developed (Fig 1). Precision recall curves were generated for each individual label as well as for the algorithm overall. We adopted a threshold value of 0.5 to yield balanced precision and recall. The AUC of the model to discriminate Group N and Group I and the confusion matrix are shown in Table 1. The total AUC was 0.95 (Group I AUC 0.48, Group N AUC 0.97) and the sensitivity, specificity, positive predictive value and negative predictive value of Group I detection were 30.8%, 97.6%, 66.7%, and 90.23%, respectively. We found that the confusing rate for Group I and Group N was 69% and 2%, respectively. Representative images from endoscopy for patients with high IBS scores (Fig 2) and high normal scores are shown (Fig 3).

### Group N vs. Groups C and D

Next, images from Group N were compared with images from endoscopy for patients in Groups C and D. As training, validation and test images, 419, 51 and 53 images, respectively,

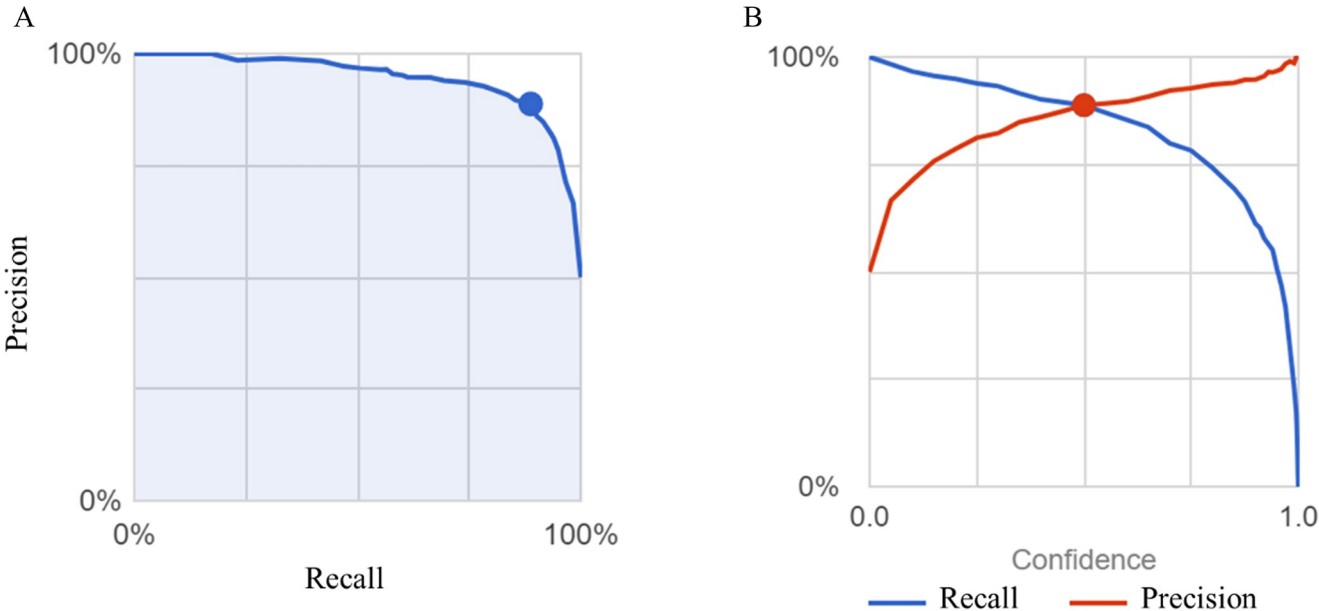

**Fig 1.** (A) The precision of the Groups I and N colon image AI model is plotted as a function of recall. The blue shaded area represents the area under the curve (AUC), and the blue dot indicates the value in the case with a reliability threshold of 0.5. (B) The intersection of the recall (blue line) and the precision (red line) is shown. The blue and red dots indicate the values when the reliability threshold is set to 0.5.

**Table 1. Groups I and N model of the colon; AUC and confusion matrix.** The confusion matrix shows how often each label was correctly classified in the model (Predictive and True labels' agreement) and the labels that were confused for that label (Predictive and True labels' disagreement).

|  | AUC |  |
|---|---|---|
| All labels | 0.95 |  |
| Group I | 0.48 |  |
| Group N | 0.97 |  |
|  | Predictive label |  |
| True label | Group I | Group N |
| Group I | 31% | 69% |
| Group N | 2% | 98% |

were used for Group C and 387, 48 and 49 images, respectively, were used for Group D. For these groups, the average precision (positive predictive value), precision, and recall of the algorithm were 83.2%, 77.71%, and 67.97%, respectively, based on automated training and testing (Fig 4). The precision recall curves and the threshold value were set as described above. The AUC of the model to discriminate among the groups and the confusion matrix are shown in Table 2. The total AUC was 0.83 (0.90 for Group N, 0.45 for Group C and 0.60 for Group D) and the sensitivity, specificity, and positive predictive value of Group N were 87.5%, 46.2%, and 79.9%, respectively. The confusing rate for Groups N, D and C was 12%, 51% and 66%, respectively.

## Group C vs. Group D

In comparing Group C and Group D, the average precision (positive predictive value), precision, and recall of the algorithm were 89.75%, 87.5%, and 87.5%, respectively, based on automated training and testing (Fig 5). The precision recall curves and the threshold value were set as described above. The AUC of the model to discriminate among the groups and the confusion matrix are shown in Table 3. The total AUC was 0.90 (0.87 for Group C and 0.94 for Group D). The confusing rate for Groups D and C was 18% and 7%, respectively. Representative images from endoscopy for patients with high IBS-D scores (Fig 6) and high IBS-C scores (Fig 7) are shown.

## Group N vs. Group I + Group C +Group D

Finally, Groups I, C, and D as a whole were merged to see if they could be distinguished from Group N using the same images. The average precision(positive predictive value), precision, and recall of the algorithm were 81.2%, 72.6%, and 72.6%, respectively.

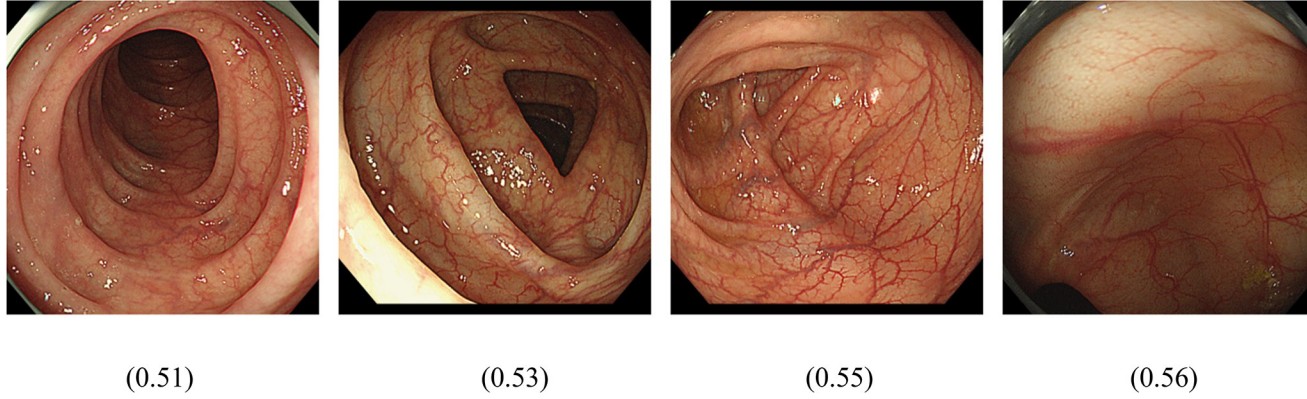

<div align="center">(0.51)  (0.53)  (0.55)  (0.56)</div>

**Fig 2. Images that scored relatively high (0 to 1) in the colon image AI model for detecting Group I are shown.** Values shown correspond to score.

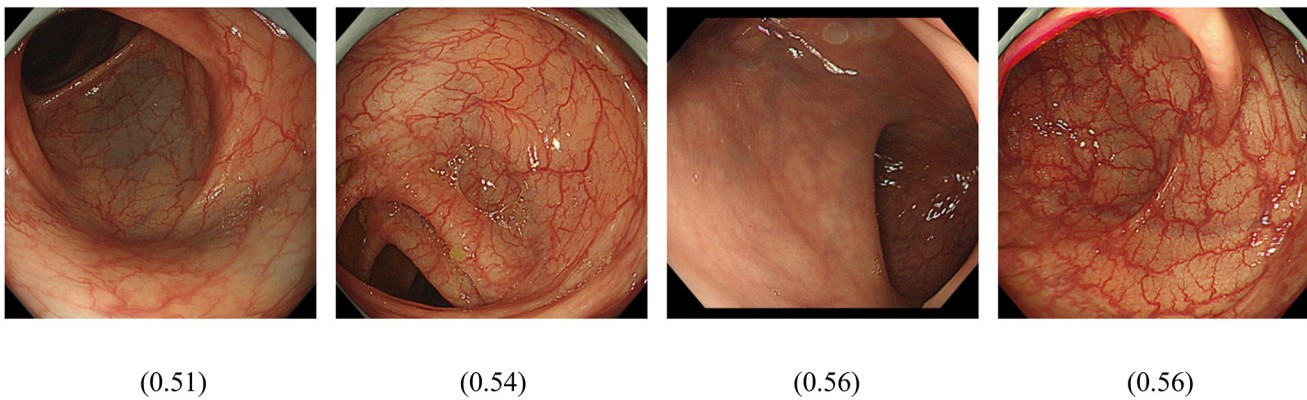

| (0.51) | (0.54) | (0.56) | (0.56) |

**Fig 3. Images that scored relatively high (0 to 1) in the colon image AI model for detecting Group N are shown.** Values shown correspond to score.

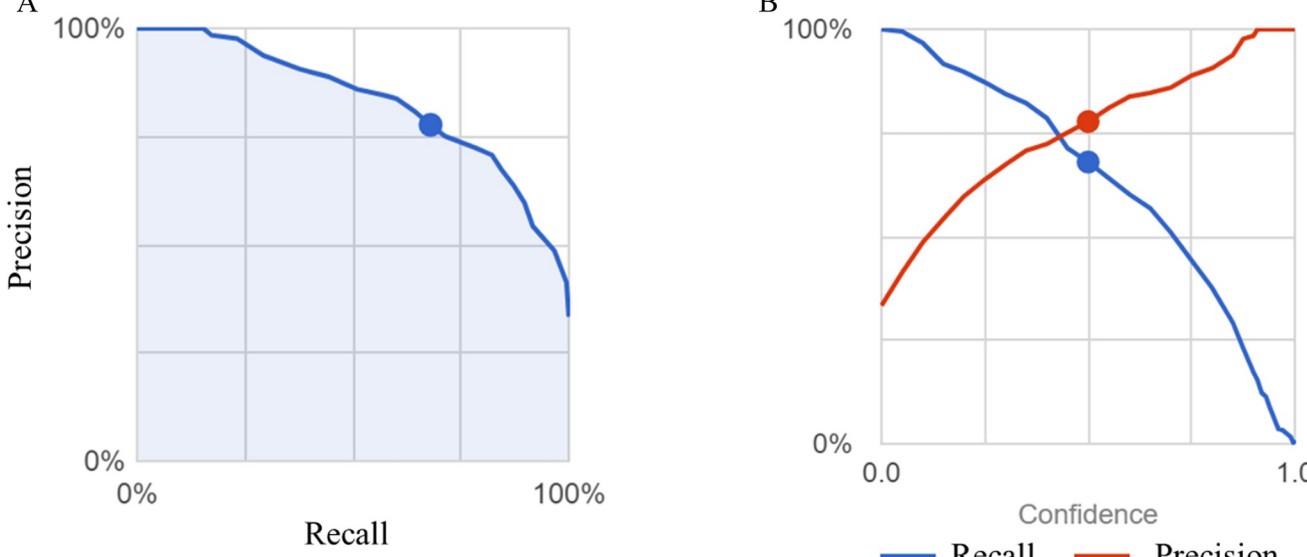

**Fig 4.** (A) The precision of the Groups N, C and D colon image AI model is plotted as a function of recall. The blue shaded area represents the area under the curve (AUC), and the blue dot indicates the value in the case with a reliability threshold of 0.5. (B) The intersection of the recall (blue line) and the precision (red line) is shown. The blue and red dots indicate the values when the reliability threshold is set to 0.5.

**Table 2. Groups N, C and D model of the colon; AUC and confusion matrix.**

|  | AUC |  |  |
|---|---|---|---|
| All labels | 0.83 |  |  |
| Group N | 0.90 |  |  |
| Group C | 0.45 |  |  |
| Group D | 0.60 |  |  |
|  | Predictive label |  |  |
| True label | Group N | Group C | Group D |
| Group N | 87% | 5% | 7% |
| Group C | 62% | 35% | 4% |
| Group D | 45% | 6% | 49% |

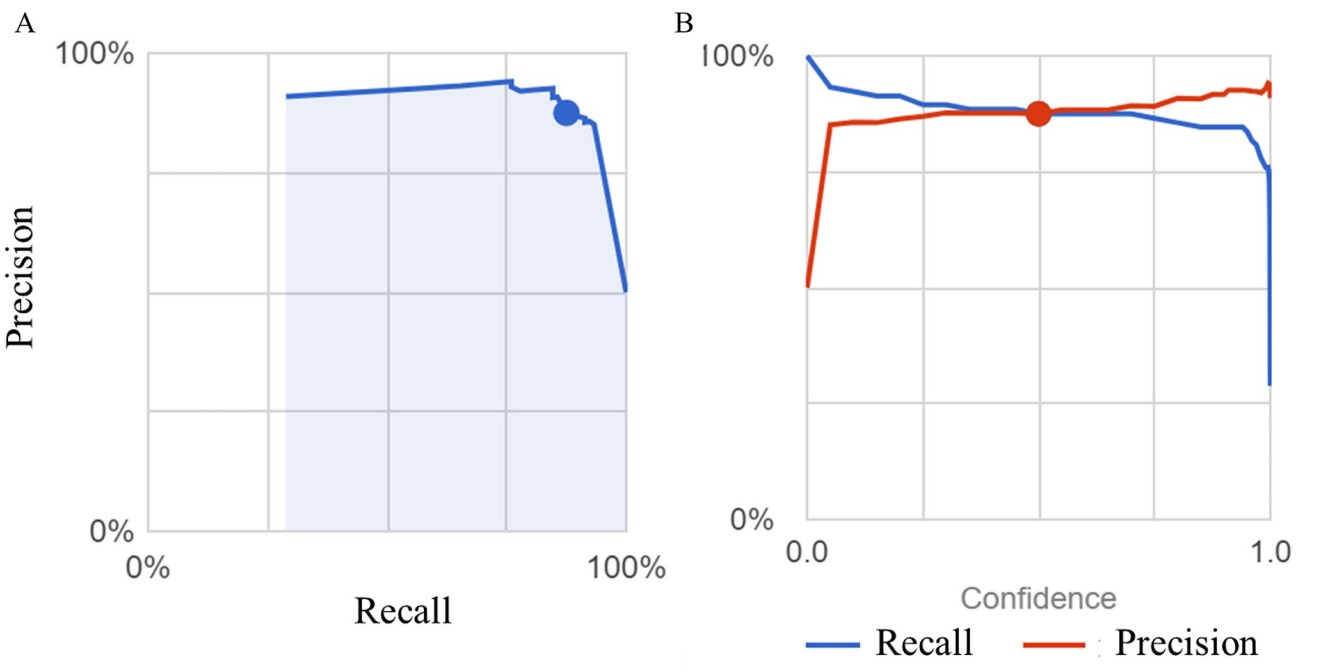

**Fig 5.** (A) The precision of the Groups C and D colon image AI model is plotted as a function of recall. The blue shaded area represents the area under the curve (AUC), and the blue dot indicates the value in the case with a reliability threshold of 0.5. (B) The intersection of the recall (blue line) and the precision (red line) is shown. The blue and red dots indicate the values when the reliability threshold is set to 0.5.

## Discussion

Endoscopic images of patients with IBS typically show no abnormalities. In this study, we investigated whether AI can detect minute endoscopic changes associated with microinflammation and other microenvironmental changes that cannot be easily detected by human observers. We constructed a code-free AI model to detect Group I with an AUC of 0.95 and a high specificity among Group N. The model was re-created separately for Groups C and D, and the AUC was slightly lower at 0.83, suggesting that Groups C and D may be distinguishable from each other. When the AI model was re-created to distinguish between the two, the AUC was 0.90, which was greater than the difference between the two groups and healthy subjects. The model for Group N vs. Group I + Group C + Group D is slightly less accurate than Group N vs. Group I, suggesting that Groups I, C, and D may be different populations in terms of images. To the best of our knowledge, this is the first AI model that can detect IBS in endoscopic images. Further investigation is needed to determine whether AI can differentially

**Table 3. Groups C and D model of the colon; AUC and confusion matrix.**

|  | AUC |  |
| --- | --- | --- |
| All labels | 0.90 |  |
| Group C | 0.87 |  |
| Group D | 0.94 |  |
|  | Predictive label |  |
| True label | Group C | Group D |
| Group C | 93% | 7% |
| Group D | 18% | 82% |

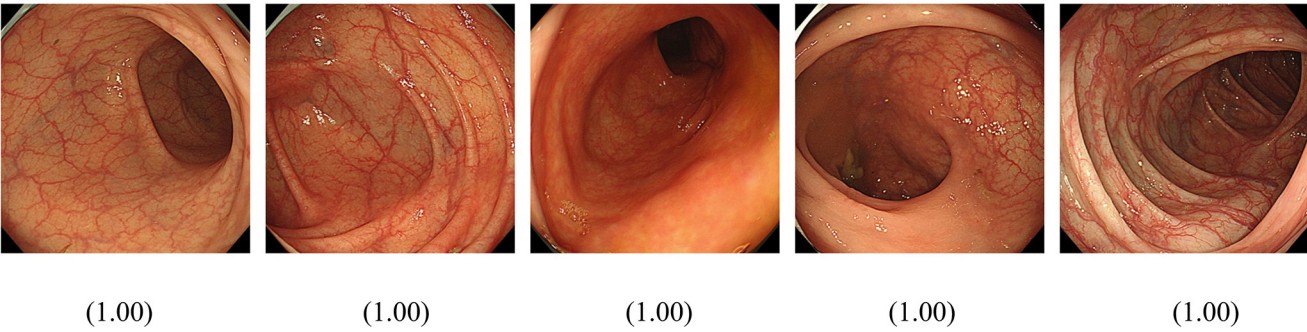

(1.00)  (1.00)  (1.00)  (1.00)  (1.00)

**Fig 6. Images that scored high (0 to 1) in the colon image AI model for detecting Group D are shown.** Values shown correspond to score.

detect histological abnormalities, the presence of biofilm, and/or deformation of the colorectal lumen. Since the model was not constructed according to segments, whether a particular segment might have a greater or lesser contribution to diagnoses was unclear. Here we assumed that any segment used to calculate the IBS score was appropriate for that segment. The advantage of this model is that it returns an IBS score independently of segment. Preliminary results showed that the greatest differences occurred in the sigmoid colon, but further detailed studies are needed before conclusions about the diagnostic value of different segments can be drawn.

This model has several limitations. First, IBS diagnosis was defined not by ROME criteria, but instead by disease names that were recorded for insurance purposes. However, ROME IV criteria are not always used in clinical practice, and thus this model is expected to be accurate because it is designed for AI use in clinical practice. Second, patients in the Groups C and D may have been treated with linaclotide and ramosetron, respectively. Third, age, gender, and treatment response were not taken into account when selecting images for the training dataset. Fourth, the minimum sample size in GCP AutoML Vision was 100 images, but ideally more than 1,000 images are required. Furthermore, since the training, validation, and test images were from the same patient population, even though very different images were used, the results require validation with an independent patient cohort. And GCP AutoML Vision does not allow iterative refinements during the training phase in the form of loss vs. epoch or accuracy vs. epoch plots, making it difficult to confirm that the model is converging on the optimal path. Last, the study groups included cases before and after treatment and cases with and without treatment response.

The prime advantage of the use of Google Cloud AutoML Vision is that it requires no coding expertise and can be easily used with datasets to build AI Models. The code-free deep learning approach used in this study has the potential to improve access of clinicians to deep

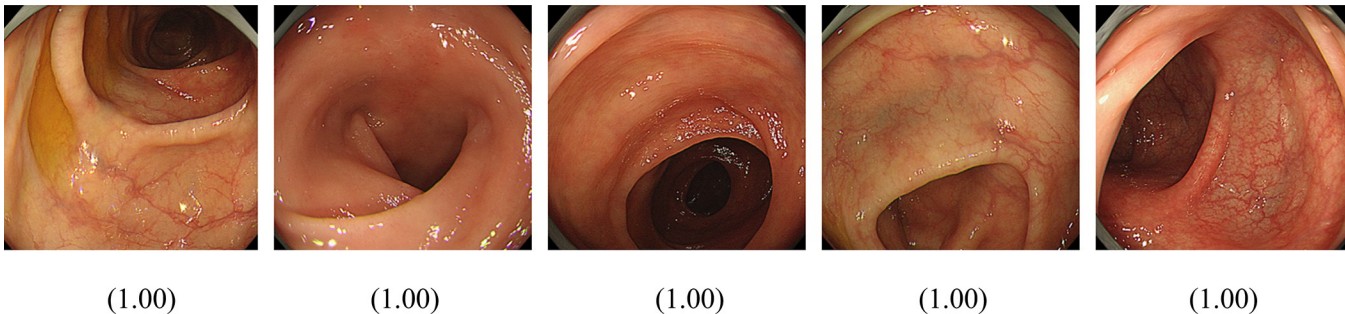

(1.00)  (1.00)  (1.00)  (1.00)  (1.00)

**Fig 7. Images that scored high (0 to 1) in the colon image AI model for detecting Group C are shown.** Values shown correspond to score.

learning [8,9]. Other research groups have already reported medical image classification and otolaryngology diagnosis that was performed with an automated, coding-free deep learning approach [10,11]. Many deep convolutional neural networks (CNN) architectures exist [13]. In 2014 the predecessor model to GCP AutoML Vision, GoogLeNet, won the ImageNet Large Scale Visual Recognition Competition (ILSVRC) an international image AI competition, and achieved high accuracy with low computational cost [14]. Later, based on the theoretical background of the Neural Architecture Search, which optimizes the neural network architecture itself, GCP AutoML enabled high-quality image classification models to be generated even by those having no expertise in machine learning. For radiological images, Resnet, which enables construction of deep neural networks with over 1,000 layers, yielded better results than those obtained with GoogLeNet [15]. Whether Microsoft Azure, which is based on Resnet, is better than GCP AutoML Vision for building code-free endoscopic image AI models is a subject for future study.

Meanwhile, the endoscopic features the Artificial Neural Network (ANN) AI classifier uses to distinguish between model patients with IBS vs. healthy controls are unclear as the current model is essentially a black box. However, an explainable AI model is becoming available [16], and by adding a function to display the areas of importance that contributed to the score, determining which endoscopic features are characteristic of IBS should be possible.

We have confirmed that AI-based algorithms are also suitable for symptom-based diagnosis. Such algorithms may be able to detect differences in endoscopic images in other functional gastrointestinal disorders, such as functional dyspepsia and nonerosive gastroesophageal reflux disease. The accuracy of the AI model for IBS is expected to vary depending on the light source setting of the endoscope and scope, and whether the same accuracy is achieved at other facilities should be explored. In summary, here we described the development of an AI model that did not require coding experience and that can differentiate Groups I, C, and D from colonoscopy images. Construction of AI models based on the presence or absence of symptoms could be a new method to diagnose functional gastrointestinal diseases.

## Acknowledgments

We thank Ayaka Maeda, Masaya Hiraki, Shun Kuraishi, and Kenji Ogawa, medical engineering technicians at the Medical Device Management Center, University of Toyama Hospital, for their support in collecting and organizing the images.

A summary of this study was presented at the 23rd Annual Meeting of the Japanese Society of Neurogastroenterology.

## Author Contributions

**Conceptualization:** Kazuhisa Tabata, Sohachi Nanjo, Iori Motoo, Takayuki Ando, Akira Teramoto, Haruka Fujinami, Ichiro Yasuda.

**Data curation:** Kazuhisa Tabata, Hiroshi Mihara, Akira Teramoto, Haruka Fujinami.

**Formal analysis:** Kazuhisa Tabata.

**Funding acquisition:** Takayuki Ando, Ichiro Yasuda.

**Investigation:** Kazuhisa Tabata.

**Methodology:** Akira Teramoto, Haruka Fujinami.

**Project administration:** Haruka Fujinami, Ichiro Yasuda.

**Resources:** Kazuhisa Tabata.

**Software:** Hiroshi Mihara.

**Supervision:** Haruka Fujinami, Ichiro Yasuda.

**Validation:** Kazuhisa Tabata, Sohachi Nanjo, Iori Motoo, Takayuki Ando, Akira Teramoto, Haruka Fujinami.

**Writing – original draft:** Kazuhisa Tabata, Hiroshi Mihara.

**Writing – review & editing:** Kazuhisa Tabata, Hiroshi Mihara, Sohachi Nanjo, Iori Motoo, Takayuki Ando, Akira Teramoto, Haruka Fujinami, Ichiro Yasuda.

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
