## [Decision Letter · Decision Letter 0]

3 Aug 2022

PDIG-D-22-00137

Artificial Intelligence Model for Analyzing Colonic Endoscopy Images to Detect Changes Associated with Irritable Bowel Syndrome

PLOS Digital Health

Dear Dr. Mihara,

Thank you for submitting your manuscript to PLOS Digital Health. After careful consideration, we feel that it has merit but does not fully meet PLOS Digital Health's publication criteria as it currently stands. Therefore, we invite you to submit a revised version of the manuscript that addresses the points raised during the review process.

Please submit your revised manuscript within 60 days Oct 02 2022 11:59PM. If you will need more time than this to complete your revisions, please reply to this message or contact the journal office at digitalhealth@plos.org. Please include the following items when submitting your revised manuscript:

We look forward to receiving your revised manuscript.

Kind regards,

Benjamin P. Geisler, M.D., M.P.H., F.A.C.P., M.R.C.P. (London), F.H.M.

Academic Editor

PLOS Digital Health

Journal Requirements:

1. In the online submission form, you indicated that “Data cannot be shared for confidentiality reasons. Queries about the data should be directed to the corresponding author.”. All PLOS journals now require all data underlying the findings described in their manuscript to be freely available to other researchers, either 1. In a public repository, 2. Within the manuscript itself, or 3. Uploaded as supplementary information.

2. Please provide separate figure files in .tif or .eps format and remove any figures embedded in your manuscript file. Please also ensure that all files are under our size limit of 10MB.

For more information about how to convert your figure files please see our guidelines: https://journals.plos.org/digitalhealth/s/figures

3. Please ensure that you refer to Table 3 in your text as, if accepted, production will need this reference to link the reader to the table.

4. All figures and supporting information files will be published under the Creative Commons Attribution License (creativecommons.org/licenses/by/4.0/). Authors retain ownership of the copyright for their article and are responsible for third-party content used in the article. 

Figures 2, 3, and 6: Please confirm (a) that you are the photographer; or (b) provide written permission from the photographer to publish the photo(s) under our CC-BY 4.0 license.

Please upload any written confirmation as an 'Other' file type. It must clarify that the copyright holder understands and agrees to the terms of the CC BY 4.0 license; general permission forms that do not specify permission to publish under the CC BY 4.0 will not be accepted. Note that uploading an email confirmation is acceptable.

Additional Editor Comments (if provided):

Reviewers' comments:

Reviewer's Responses to Questions

**Comments to the Author**

1. Does this manuscript meet PLOS Digital Health’s publication criteria? Is the manuscript technically sound, and do the data support the conclusions? The manuscript must describe methodologically and ethically rigorous research with conclusions that are appropriately drawn based on the data presented.

Reviewer #1: Yes

Reviewer #2: Yes

Reviewer #3: Partly

Reviewer #4: Yes

2. Has the statistical analysis been performed appropriately and rigorously?

Reviewer #1: Yes

Reviewer #2: Yes

Reviewer #3: Yes

Reviewer #4: No

3. Have the authors made all data underlying the findings in their manuscript fully available (please refer to the Data Availability Statement at the start of the manuscript PDF file)?

Reviewer #1: Yes

Reviewer #2: Yes

Reviewer #3: No

Reviewer #4: Yes

4. Is the manuscript presented in an intelligible fashion and written in standard English?

Reviewer #1: Yes

Reviewer #2: Yes

Reviewer #3: Yes

Reviewer #4: Yes

5. Review Comments to the Author

Reviewer #1: The authors’ purpose was to determine whether image AI models can differentiate between different types of irritable bowel syndrome (IBS) and healthy colonoscopic images using Google cloud AutoML Vision. 

I have some comments to make on their research.

1. Page 6, line 90-92: I think that that the text and references do not match. The authors should check this.

2. Page 7, line 98: The content of the reference and the text appear not to match.

3. Page 7, line 98-100: The content of the references clearly does not match the text.

4. Page 8, line120-122: The authors mention some groups of patients. However, they do not give any background information for them. The authors should state such background factors as gender, age, history of illness, and treatment history. In addition, I think that the total numbers of patients are not sufficient to make an AI model.

Reviewer #2: In this retrospective study from Japan, Mihara et al used an artificial neural network (ANN) AI classifiers to analyze and model endoscopic images from colonoscopies to differentiate between patients with various irritable bowel syndrome/IBS subtypes (IBS, IBS-D, IBS-C) versus healthy controls with relatively good performance with an area under the curve (AUC) ranging from 0.85-0.93. Objective markers of diagnosing IBS are much needed. This is an interesting study and addresses an important topic. I have the following critiques and recommendations: 

1. Abstract, Background/Aims: "IBS is not an organic disease, and the patients typically show no abnormalities in lower gastrointestinal endoscopy." The authors are correct that IBS patients usually have normal colonoscopies (endoscopic and histologic). Since most colonoscopies are normal in patients with IBS, the authors should clarify why they chose to review colonoscopy images by AI modeling.

2. Abstract, Background/Aims: "Recently, biofilm formation has been visualized by endoscopy, and the ability of endoscopy to detect microscopic changes due to dysbiosis and microinflammation has been reported." If this is the case, why did the authors only look at patients with IBS without biofilms? Why mention biofilms if their study doesn't even measure this?

3. Methods: "These names included "Irritable bowel syndrome (group I)." The authors should clarify what type of IBS patients were included in Group 1 and how they differed from the other groups with constipation-predominant (IBS-C) and diarrhea-predominant (IBS-D).

4. Methods: Among patients with IBS-D, were other etiologies of chronic diarrhea ruled out (e.g. inflammatory bowel disease, microscopic colitis, small intestinal bacterial overgrowth)?

5. Methods: The authors should clarify the rationale for colonoscopy in the included patients? Was this screening colonoscopy for colon cancer? Was colonoscopy done as part of workup of altered bowel habits (e.g. diarrhea)?

6. Results: The authors should include a Table 1 with baseline clinical characteristics of included patients and healthy controls include patient age, sex, medications, etc. 

7. Methods: The authors should expand the details of the AI classifier they used for the study. Was was ANN used versus other AI classifiers? What are the benefits of this strategy when analyzing endoscopic images?

8. Methods/Results: For the Figures, the author state "High-scoring images in patients with IBS (score 0-1)." What do these scores refer to? How was this derived and what do they mean clinically?

9. Methods/Results: What endoscopic features (e.g. alterations in vascular patterns, mucosal folds, colonic mucosa patterns or color, presence or scarring or erythema) did the artificial neural network (ANN) AI classifier use to differentiate/model patients with IBS vs healthy controls? 

10. Methods/Results: Older patients with chronic constipation often have diverticulosis across the colon. How did the AAN AI classifier account for diverticulosis in the predictive model? Did diverticulosis predict IBS-C? Similarly, patients who use laxatives sometimes have discoloration of the colon (melanosis coli). How did the AAN AI classifier model this finding?

11. Methods/Results: Were biopsies obtained from the included patient and histologic assessment performed to rule out microscopic colitis? Sometimes during colonoscopy, colonic mucosa appears endoscopically normally, but has inflammation at the histologic level. 

12. Methods/Results: Different segments of the colon (right colon, transverse, rectum, etc) have different endoscopic features. For all the included patients, the authors should clarify how many endoscopic images were obtained for each segment and clarify whether images from specific segments were more diagnostic/contributed more to AI-derived model to differentiate IBS patients from healthy controls.

13. Methods/Results: The analysis could be more robust and not only differentiate/classify IBS vs non-IBS patients, but also should correlate the endoscopic images with validated IBS severity scores (e.g. Birmingham IBS Symptom Questionnaire , IBS Symptom Severity Scale, etc).

14. Methods/Discussion: The authors should state that the small sample size and lack of validation of the ANN AI model in an independent cohort of patients are additional limitations of the study.

15. Discussion: The authors state "Endoscopic images of IBS typically show no abnormalities, but we investigated whether AI could detect microinflammation that cannot be easily detected by human observers." How was microinflammation defined and assessed from colonoscopy endosocopic images in the included cohort of patients?

Reviewer #3: The authors applied the Google cloud platform AutoML Vision to build a model to differentiate between patients with IBS and asymptomatic healthy controls based on colonoscopy images. The validation cohort was a subset of the training set and the model could differentiate asymptomatic controls and IBS patients with a sensitivity of 30 % and specificity of 97 %.

Before publication some major points need to be clarified:

- The “group I” comprised of “IBS” patients while “group C” and “D” comprised of IBS-C and IBS-D patients. While IBS-D and IBS-C are clear subgroups, it is unclear, what defines group “I”. Also the diagnostic algorithm for IBS diagnosis is not well defined. Rome criteria should be applied. For the sake of clinical relevance we recommend combining IBS-M and IBS-D and excluding IBS-C from such analysis. Also a larger dataset (>100 patients and controls) would reduce overfitting. 

- The authors show colonoscopy pictures of high scoring IBS and control patients (Figure 2 and 3). There is no evident visual difference between those pictures, but authors suggest that a machine learning model might pick up differences in i.e. vasculation pattern. In fact endoscopically visible biofilms as mentioned in their introduction are obviously not present in the representative images. Were any biofilms detectable in this study cohort ? Were images with incomplete bowel cleansing removed from the data set ? Then they likely have remove obvious biofilm patients from their data set.

-Machine learning is highly sensitive to bias. Authors need to improve on cohort characterization; were all patients scoped with the same model of endoscope. Where patients from each cohort scoped in the same hospital/ by the same endoscopist?

Minor points:

-Abstract, Background and Aims: authors start with “IBS is not an organic disease...” but continue to elaborate on biofilm formation, dysbiosis and microinflammation. This mismatch needs to be elaborated.

Reviewer #4: The authors made a image recognition model using Google AutoML vision. They successfully distinguished healthy vs IBD samples. 

The approach has merits and significant advantage in understanding the pathophysiology of IBD.

However, the authors are using Google AutoML tool as a black box. The training and validation statistics are only limited

the to final ROC or contingency tables.

I think the authors should reveal how training improved the models in different iterations. They should clearly separate traning and validation datasets with number of samples.

The tables with AUC should also reveal number of samples used. 

The authors didn't reveal how the images were scores and how were they ranked.

6. PLOS authors have the option to publish the peer review history of their article (what does this mean?). If published, this will include your full peer review and any attached files.

**Do you want your identity to be public for this peer review?** For information about this choice, including consent withdrawal, please see our Privacy Policy.

Reviewer #1: No

Reviewer #2: Yes: John Gubatan, MD

Reviewer #3: No

Reviewer #4: Yes: Debashis Sahoo

---

## [Decision Letter · Decision Letter 1]

24 Oct 2022

PDIG-D-22-00137R1

Artificial Intelligence Model for Analyzing Colonic Endoscopy Images to Detect Changes Associated with Irritable Bowel Syndrome

PLOS Digital Health

Dear Dr. Mihara,

Thank you for submitting your manuscript to PLOS Digital Health. After careful consideration, we feel that it has merit but does not fully meet PLOS Digital Health's publication criteria as it currently stands. Therefore, we invite you to submit a revised version of the manuscript that addresses the points raised during the review process.

Please perform an additional analysis of a combined group IBS patients vs. controls, as suggested by one of the reviewers during the first revision cycle. Please see reviewer #3's comment regarding this below.

Please submit your revised manuscript within 60 days Dec 23 2022 11:59PM. If you will need more time than this to complete your revisions, please reply to this message or contact the journal office at digitalhealth@plos.org. Please include the following items when submitting your revised manuscript:

We look forward to receiving your revised manuscript.

Kind regards,

Benjamin P. Geisler, M.D., M.P.H., F.A.C.P., M.R.C.P. (London), F.H.M.

Academic Editor

PLOS Digital Health

Journal Requirements:

Additional Editor Comments (if provided):

Please perform an additional analysis of a combined group IBS patients vs. controls, as suggested by one of the reviewers during the first revision cycle.

Reviewers' comments:

Reviewer's Responses to Questions

**Comments to the Author**

1. If the authors have adequately addressed your comments raised in a previous round of review and you feel that this manuscript is now acceptable for publication, you may indicate that here to bypass the “Comments to the Author” section, enter your conflict of interest statement in the “Confidential to Editor” section, and submit your "Accept" recommendation.

Reviewer #1: All comments have been addressed

Reviewer #2: All comments have been addressed

Reviewer #3: (No Response)

Reviewer #4: (No Response)

2. Does this manuscript meet PLOS Digital Health’s publication criteria? Is the manuscript technically sound, and do the data support the conclusions? The manuscript must describe methodologically and ethically rigorous research with conclusions that are appropriately drawn based on the data presented.

Reviewer #1: Yes

Reviewer #2: Yes

Reviewer #3: No

Reviewer #4: Yes

3. Has the statistical analysis been performed appropriately and rigorously?

Reviewer #1: Yes

Reviewer #2: Yes

Reviewer #3: Yes

Reviewer #4: No

4. Have the authors made all data underlying the findings in their manuscript fully available (please refer to the Data Availability Statement at the start of the manuscript PDF file)?

Reviewer #1: Yes

Reviewer #2: Yes

Reviewer #3: No

Reviewer #4: Yes

5. Is the manuscript presented in an intelligible fashion and written in standard English?

Reviewer #1: Yes

Reviewer #2: Yes

Reviewer #3: Yes

Reviewer #4: Yes

6. Review Comments to the Author

Reviewer #1: This revised paper is well written and acceptable to be published in PLOS Digithal Health.

Reviewer #2: The authors have addressed my critiques to the best of their available datasets.

Reviewer #3: The authors did not perform the requested analysis of all IBS patients combined against controls. Splitting IBS patients into small ill-defined groups and comparing against a bigger cohort increases the possibility of overfitting. In terms of study design neither image location, endoscopic equipment nor disease classification was standardized. Furthermore, the authors could not specify relevant changes such as endoscopically visible biofilms in their images. An independent control cohort, which underlines clinical applicability of the model is lacking. It is thus more likely that the model is able to detec a signal for individual patients instead of underlying features of IBS.

Reviewer #4: The authors have not addressed my question of providing iterative improvements of the ML models during training phase. This data is crucial because it will show if the models are converging and achieving optimal path or no change in improvements at all. This is usually done by showing a plot of Loss vs Epochs or Accuracy vs Epochs.

7. PLOS authors have the option to publish the peer review history of their article (what does this mean?). If published, this will include your full peer review and any attached files.

**Do you want your identity to be public for this peer review?** For information about this choice, including consent withdrawal, please see our Privacy Policy. 

Reviewer #1: No

Reviewer #2: Yes: John Gubatan

Reviewer #3: No

Reviewer #4: Yes: Debashis Sahoo

---

## [Decision Letter · Decision Letter 2]

12 Jan 2023

Artificial Intelligence Model for Analyzing Colonic Endoscopy Images to Detect Changes Associated with Irritable Bowel Syndrome

PDIG-D-22-00137R2

Dear Hiroshi Mihara,

We are pleased to inform you that your manuscript 'Artificial Intelligence Model for Analyzing Colonic Endoscopy Images to Detect Changes Associated with Irritable Bowel Syndrome' has been provisionally accepted for publication in PLOS Digital Health.

Best regards,

Benjamin P. Geisler, M.D., M.P.H., F.A.C.P., M.R.C.P. (London), F.H.M.

Academic Editor

PLOS Digital Health

Reviewer Comments (if any, and for reference):

Reviewer's Responses to Questions

**Comments to the Author**

1. If the authors have adequately addressed your comments raised in a previous round of review and you feel that this manuscript is now acceptable for publication, you may indicate that here to bypass the “Comments to the Author” section, enter your conflict of interest statement in the “Confidential to Editor” section, and submit your "Accept" recommendation.

Reviewer #2: All comments have been addressed

Reviewer #4: All comments have been addressed

2. Does this manuscript meet PLOS Digital Health’s publication criteria? Is the manuscript technically sound, and do the data support the conclusions? The manuscript must describe methodologically and ethically rigorous research with conclusions that are appropriately drawn based on the data presented.

Reviewer #2: Yes

Reviewer #4: Yes

3. Has the statistical analysis been performed appropriately and rigorously?

Reviewer #2: Yes

Reviewer #4: Yes

4. Have the authors made all data underlying the findings in their manuscript fully available (please refer to the Data Availability Statement at the start of the manuscript PDF file)?

Reviewer #2: Yes

Reviewer #4: Yes

5. Is the manuscript presented in an intelligible fashion and written in standard English?

Reviewer #2: Yes

Reviewer #4: Yes

6. Review Comments to the Author

Reviewer #2: (No Response)

Reviewer #4: All concerns are addressed.

7. PLOS authors have the option to publish the peer review history of their article (what does this mean?). If published, this will include your full peer review and any attached files.

**Do you want your identity to be public for this peer review?** For information about this choice, including consent withdrawal, please see our Privacy Policy.

Reviewer #2: **Yes: **John Gubatan, MD

Reviewer #4: **Yes: **Debashis Sahoo
